From promise to practice: pairing non-invasive sampling with genomics in conservation

Russello Michael A. 1 michael.russello@ubc.ca
Waterhouse Matthew D. 1
Etter Paul D. 2
Johnson Eric A. 2
1 Department of Biology, University of British Columbia , Kelowna, BC , Canada
2 SNPsaurus , Eugene, OR , USA
Fonseca Dina
Electronic publication date: 2015 Jul 21
Publication date: 2015
Volume: 3
Electronic Location ID: e1106
Received 2015 Apr 20; Accepted 2015 Jun 25
Copyright: © 2015 Russello et al.
Copyright year: 2015
Copyright holder: Russello et al.
License: This is an open access article distributed under the terms of the Creative Commons Attribution License, which permits unrestricted use, distribution, reproduction and adaptation in any medium and for any purpose provided that it is properly attributed. For attribution, the original author(s), title, publication source (PeerJ) and either DOI or URL of the article must be cited.
License URL: https://creativecommons.org/licenses/by/4.0/

Keywords: Genotyping-by-sequencing, Restriction-site associated DNA sequencing, Single nucleotide polymorphism, Ochotona princeps, Hair, Biodiversity, Climate change, American pika, NextRAD

Funding: Sarah and Daniel Hrdy Fellowship in Conservation Biology Natural Sciences and Engineering Research Council of Canada #341711 Seattle City Lights #2013-03 Funding for this work was provided by the Sarah and Daniel Hrdy Fellowship in Conservation Biology (Harvard University to MR), the Natural Sciences and Engineering Research Council of Canada (Discovery Grant # 341711 to MR) and Seattle City Lights (grant # 2013-03 to MR). The funders had no role in study design, data collection and analysis, decision to publish, or preparation of the manuscript.

==============================
Conservation genomics has become an increasingly popular term, yet it remains unclear whether the non-invasive sampling that is essential for many conservation-related studies is compatible with the minimum requirements for harnessing next-generation sequencing technologies. Here, we evaluated the feasibility of using genotyping-by-sequencing of non-invasively collected hair samples to simultaneously identify and genotype single nucleotide polymorphisms (SNPs) in a climate-sensitive mammal, the American pika (Ochotona princeps). We identified and genotyped 3,803 high-confidence SNPs across eight sites distributed along two elevational transects using starting DNA amounts as low as 1 ng. Fifty-five outlier loci were detected as candidate gene regions under divergent selection, constituting potential targets for future validation. Genome-wide estimates of gene diversity significantly and positively correlated with elevation across both transects, with all low elevation sites exhibiting significant heterozygote deficit likely due to inbreeding. More broadly, our results highlight a range of issues that must be considered when pairing genomic data collection with non-invasive sampling, particularly related to field sampling protocols for minimizing exogenous DNA, data collection strategies and quality control steps for enhancing target organism yield, and analytical approaches for maximizing cost-effectiveness and information content of recovered genomic data.

Introduction

There has been much discussion on the transition of conservation genetics to conservation genomics (Helyar et al., 2011; McMahon, Teeling & Hoglund, 2014). Genomic analysis provides the advantage of assessing natural selection and adaptive genetic variation (Schoville et al., 2012), accurately estimating levels of genome wide diversity (Vali et al., 2008), and providing novel information for delineating conservation units (Funk et al., 2012) and informing management strategies (Hoffmann et al., 2015). Yet, it remains unclear the degree to which the conservation community as a whole has embraced genomics as a useful tool, suggesting significant gaps in methodology and analysis that must be overcome before the technology is ready for real-world applications (Shafer et al., 2015). One methodological aspect that has yet to be formally considered is the sample source of DNA; many population genetic studies of elusive or endangered species must rely on non-invasively collected samples. There is an expansive literature on the use of DNA from hair, feces, feathers and other non-invasively sampled materials for investigating the ecology, behavior, and population history of wildlife species (reviewed in Beja-Pereira et al., 2009; Waits & Paetkau, 2005). These studies have largely been based on single locus mitochondrial DNA (mtDNA) sequencing or multi-locus nuclear genotyping of hyper-variable loci, such as microsatellites (Taberlet, Waits & Luikart, 1999; Waits & Paetkau, 2005). Modern genotyping-by-sequencing approaches, such as those that rely on restriction-site associated DNA (RAD) tags, typically call for 1 µg of high quality DNA for library construction (Baird et al., 2008; Etter et al., 2011). However, even highly refined DNA extraction protocols from non-invasively collected starting materials typically yield low concentrations of DNA, which may also contain PCR inhibitors (Beja-Pereira et al., 2009; Roon, Waits & Kendall, 2003; Smith & Wang, 2014; Taberlet, Waits & Luikart, 1999; Waits & Paetkau, 2005). To date, it remains unclear whether the non-invasive sampling that is essential for many conservation-related studies is compatible with the minimum requirements for harnessing the next-generation sequencing (NGS) technologies necessary for expanding conservation genetics in the genomics era.

The American pika, Ochotona princeps, is an example of an elusive species that has benefited from the pairing of non-invasive sampling with genetic data collection. A small lagomorph, the American pika is discontinuously distributed in mountainous areas throughout western North America from central British Columbia and Alberta, Canada, south to the Sierra Nevada in California and east to New Mexico, USA. Pikas are restricted to talus slopes in proximity to meadows that provide their food (Smith & Weston, 1990). Exhibiting highly nonrandom distributions across mountaintop habitats, the average elevation of Great Basin O. princeps populations is currently ∼783 m higher than during the late Wisconsinan (Grayson, 2005). In general, lower elevational limits are constrained by an inability to tolerate high temperatures, while high altitude distribution is possible through adaptation to hypoxic environments. The fragmented nature of their habitats has propelled O. princeps as a focal mammalian species for studies of metapopulation dynamics, island biogeography, source–sink dynamics (Beever et al., 2013; Peacock & Smith, 1997a), and extinction risk in the face of climate change (Beever et al., 2010; Hafner, 1993; Smith, 1974; Stewart et al., 2015).

Recent genetic studies of American pika have relied on samples obtained non-invasively using hair snares, which have greatly enhanced sample sizes while minimizing sampling effort (Henry & Russello, 2011). These studies revealed restricted dispersal capacity (Henry, Sim & Russello, 2012) and preliminary evidence for adaptive population divergence of American pika along elevation gradients at their northern range margin (Henry & Russello, 2013). These latter findings were based on amplified fragment length polymorphism (AFLP)-based genomic scans. In addition to other undesirable properties, AFLPs are anonymous, dominant markers, which precluded the identification of genes responsible for the observed adaptive divergence. Single nucleotide polymorphisms (SNPs), with their broad genomic coverage and better understood mutation models, would overcome many of these limitations if they can be effectively genotyped within the constraints imposed by this system and others involving elusive and endangered species.

In the current study, we used nextRAD (Nextera-tagmented, reductively-amplified DNA) genotyping to collect SNP data from American pikas sampled along parallel elevational gradients to: (1) evaluate the feasibility of using DNA from non-invasively collected hair samples to simultaneously identify and genotype SNPs in an elusive species; and (2) provide preliminary insights into patterns of neutral and adaptive population divergence within this system.

Materials and Methods

Sample collection

This study was conducted in the North Cascades National Park, Washington, USA (Fig. 1). Sites within this national park present the opportunity to sample American pika along steep elevational transects where climates change rapidly over short linear distances, while controlling for other environmental and historical factors. Additionally, while pika are currently abundant in the park, this area has been disproportionally affected by climate change (Karl et al., 2009). Pika populations were sampled along two elevational transects (Pyramid Peak (PP) and Thornton Lakes (TL)) between July and August 2013. Sites within transects ranged from 450 m to 1,700 m, representing an approximate 6 °C gradient in mean annual temperature (Briggs et al., 1997) over less than 6.5 km distance (Fig. 1).

Figure 1 Sites in North Cascades National Park, Washington, USA where America pika hair samples were non-invasively collected.

Topographic lines represent 100 m elevation. Inset shows a Structure bar plot depicting the model-based clustering results for all sites within the Pyramid Peak (PP) and Thornton Lake (TL) elevational transects based on 3,748 neutral single nucleotide polymorphisms.

Non-invasive snares were used to obtain hair samples from 12 individuals at four sites along each of the two transects (n = 96) following Henry & Russello (2011). To minimize resampling the same animal, snares were set a minimum of 15 m apart and only one sample from each snare was used. Subsequent genetic data were used to test the assumption that each sample possessed a unique genotype (see below). All samples were collected under United States Department of Interior National Park Service permit #NOCA-2014-SCI-0022 and in accordance with animal care protocol (A11-0371) as approved by the University of British Columbia’s Animal Care & Biosafety Committee.

DNA isolation, genomic data collection and SNP discovery

Total genomic DNA was extracted using the DNA IQ Tissue and Hair Extraction Kit (Promega, Madison, WI, USA) following the manufacturer’s protocol. Each sample contained 60 hair follicles with the majority of the hair shaft removed under a dissecting microscope to reduce protein and other contamination. All DNA extractions were conducted in a separate laboratory free of concentrated PCR products. Negative controls were included in each extraction to monitor contamination. DNA quantifications were conducted using real-time PCR fluorescence measurements of double stranded DNA (Blotta et al., 2005) and the Quant-it kit (Life Technologies, Foster City, CA).

Genomic DNA was converted into nextRAD genotyping-by-sequencing libraries (SNPsaurus, LLC). The nextRAD method uses selective PCR primers to amplify genomic loci consistently between samples. Genomic DNA (10 ng or less depending upon extraction yield) was first fragmented with Nextera reagent (Illumina, Inc), which also ligates short adapter sequences to the ends of the fragments. Fragmented DNA was then amplified, with one of the primers matching the adapter and extending 9 arbitrary nucleotides into the genomic DNA with the selective sequence. Thus, only fragments starting with a sequence that can be hybridized by the selective sequence of the primer will be efficiently amplified. The resulting fragments are fixed at the selective end, and have random lengths depending on the initial Nextera fragmentation. Because of this, amplified DNA from a particular locus is present at many different sizes and careful size selection of the library is not needed. For this project, an arbitrary 9-mer was chosen from those previously validated in smaller genomes, which did not appear to target repeat-masked regions in the publically available American pika genomic scaffolds (Ensembl, release 74, Ochotona_princeps.74.dna_sm.toplevel.fa) and that would approximate the results of standard RAD sequencing projects using SbfI (Baird et al., 2008).

Since these samples were collected non-invasively, it was important to assess the proportion of sequence reads in each sample that originated from the target organism relative to other environmental sources prior to conducting genotyping analysis. This was done using a custom script (SNPsaurus, LLC) that randomly sampled 1,000 high-quality reads from each sample and aligned those to the publically available American pika genomic scaffolds as well as subjected them to a blastn (Altschul et al., 1997) search of all sequences in the NCBI non-redundant database. Only samples that had greater than 50% sequencing reads that mapped to Ochotona princeps were retained for genotyping analysis.

The genotyping analysis used custom scripts (SNPsaurus, LLC) that created a reference from abundant reads present between 500 and 2,000 times across the combined set of samples, mapping all of the reads to the reference allowing two mismatches. The identified variants were then filtered by removing loci that had more than the expected maximum of two alleles and those that were present in less than 10% of all samples.

Following assembly, mapping and variant detection, the data were further filtered to maximize data quality. We retained only those loci that were genotyped in ≥50% of individuals from each transect, had a minor allele frequency ≥0.05, and a minimum coverage of 6X for homozygotes (affording 95% confidence in the genotype) while heterozygotes were required to have a minimum of 2X coverage per allele for each individual. These values were chosen to minimize null alleles and sequencing errors from biasing homozygote and heterozygote genotype calls, respectively. We then removed loci that displayed significant deviation from Hardy-Weinberg equilibrium (HWE) in more than two sites per transect as assessed using the method of Guo & Thompson (1992) as implemented in Genepop 4.3 (Raymond & Rousset, 1995; Rousset, 2008).

To ensure that only non-redundant samples were included in subsequent analyses, we conducted genotype matching across a random subset of 100 loci. We conducted the match analysis and calculated the multi-locus probability of identity (Waits, Luikart & Taberlet, 2001) for the 100 randomly chosen loci using GenAlEx (Peakall & Smouse, 2006). Only samples with unique genotypes were retained.

Outlier locus detection and annotation

Polymorphic loci were screened for statistical outliers using the Bayesian simulation method of Beaumont & Balding (2004) as implemented in Bayescan 2.1 (Foll & Gaggiotti, 2008). This analysis was run independently for each transect, with all samples coded by site (PP1-PP4, TL1-TL4). We used a prior odds value of 10, with 100,000 iterations and a burn-in of 50,000 iterations. We identified loci that were significant outliers at a q-value of 0.20. A q-value is a false discovery rate (FDR) analogue of the p-value, with the former only defined in the context of multiple testing, whereas the latter is defined on a single test. Consequently, a 20% threshold for q-values is much more stringent than a 20% threshold for p-values in classical statistics. To test for non-random association of genotypes, linkage disequilibrium was assessed between all pairs of outlier loci in each population using the exact test of Guo & Thompson (1992) and 10,000 dememorization steps, 100 batches, and 10,000 iterations per batch as implemented in Genepop 4.3 (Raymond & Rousset, 1995; Rousset, 2008). In addition, each haplotype from all nextRAD-tags that contained outlier loci were subject to a blastn (Altschul et al., 1997) search of all sequences in the NCBI non-redundant database (word size = 11; mismatch scores = 2, −3; maximum e-value = 10–15). To reduce annotations to repetitive sequences in the database, we required either a unique blastn hit or a top hit with an e-value that was at least an order of magnitude lower than the next closest hit.

Population genetic analyses

We segregated loci into two datasets including: (1) all loci identified as an outlier (“outlier dataset”); and (2) all loci not identified as an outlier (“neutral dataset”). The neutral dataset was used to conduct standard population genetic analyses for quantifying the extent and distribution of variation within and among sites. Within sites, proportion of polymorphic loci, observed (Ho) and expected (He) heterozygosity, and gene diversity (Ng) were calculated using Arlequin 3.5 (Excoffier & Lischer, 2010). Global tests for heterozygote deficit were conducted using Fisher’s method and 10,000 dememorization steps, 100 batches, and 10,000 iterations per batch as implemented in Genepop 4.3 (Raymond & Rousset, 1995; Rousset, 2008). The inbreeding coefficient, Fis, was calculated for each site as implemented in Genetix (Belkhir et al., 2004), with significance assessed using 1,000 permutations. To evaluate whether site-level genetic diversity was correlated with elevation and sample size, we conducted linear regression analyses implemented in R v. 3.1 (R Development Core Team, 2011).

Levels of genetic differentiation among groups were estimated by pairwise comparisons of θ (Weir & Cockerham, 1984), as calculated in Genetix (Belkhir et al., 2004), and evaluated using 1,000 permutations. The hierarchical organization of genetic variation within and among transects was calculated using an analysis of molecular variance (amova) as implemented in Arlequin 3.5 (Excoffier & Lischer, 2010), with significance assessed using 1,000 permutations. In addition, the model-based clustering method implemented in Structure 2.3.4 (Pritchard, Stephens & Donnelly, 2000) was used to infer the number of discrete genetic units across both transects. Run length was set to 100,000 MCMC replicates after a burn-in period of 100,000 using correlated allele frequencies under a straight admixture model. We varied the number of clusters (K) from 1 to 10, with 10 replicates for each value of K. The most likely number of clusters was determined by plotting the log probability of the data (ln Pr(X|K)) (Pritchard, Stephens & Donnelly, 2000) across the range of K values tested and selecting the K where the value of ln Pr(X|K) plateaued as suggested in the Structure manual. We also employed the ΔK method (Evanno, Regnaut & Goudet, 2005) as calculated in Structure Harvester (Earl, 2011). Results for the identified optimal values of K were summarized using clumpp (Jakobsson & Rosenberg, 2007) and plotted using distruct (Rosenberg, 2004). In order to test for unrecognized substructure in the broader Structure analysis, we repeated the above analysis for each transect separately using neutral and outlier loci.

Results

Data quality

The mean starting DNA concentration recovered from the non-invasively collected hair samples was 0.55 ng/µl with as little as 1 ng total for some samples. The mean number of sequencing reads per sample was 1,863,634. Ten samples yielded less than 100,000 sequencing reads, likely due to the degraded quality and very low quantity of starting DNA. Nineteen additional samples had less than 50% of their sequencing reads mapping to O. princeps. Sixteen of these samples had high proportions of sequence reads matching with two small mammals that likely co-occur in the sampling area (Mus musculus (n = 13) and Spermophilus (n = 3)), with others matching Homo sapiens (n = 2) and Zea mays (n = 1). The above samples (n = 29) with low overall sequence reads or a low proportion mapping to O. princeps (or both) were removed, leaving 67 samples from eight sites across two elevational transects that were subject to all downstream analyses (Fig. 1 and Table 2).

We identified 9,825 SNPs that met the minimum parameters for recovering genotypes. To minimize linkage, we retained the highest coverage SNP from each contig, resulting in 3,830 SNPs. Twenty-seven SNPs deviated from HWE in two or more sites per transect and were removed from the dataset. Consequently, all downstream analyses were based on genotypic data at 3,803 SNPs. All 67 of the retained samples possessed unique genotypes at a random subset of 100 loci (average probability of identity within each sampling site = 1.1 × 10−23), suggestive of unique individuals.

Outlier detection and annotation

Outlier detection identified 37 loci along the TL transect and 18 loci along the PP transect, none of which were shared. There was no evidence of significant deviation from linkage equilibrium for any pairwise comparison of outlier loci across populations. Fourteen outlier loci unambiguously matched sequences from the NCBI nr database, five of which annotated to genes of known functions (Table 1). Locus 57863_76 identified from the PP transect mapped to the receptor tyrosine kinase-like orphan receptor 2 (ROR2) gene that is part of a conserved family that function in developmental processes including skeletal and neuronal development, cell movement and cell polarity (Green, Kuntz & Sternberg, 2008). Likewise, locus 108547_114 identified from the TL transect annotated to another gene encoding a cell surface tyrosine kinase receptor (beta-type platelet-derived growth factor receptor), but for members of the platelet-derived growth factor family (Shim et al., 2010). Locus 28594_45 was similar to the laminin alpha 3 gene in humans that codes for a protein that is essential for formation and function of the basement membrane, with additional functions in regulating cell migration and mechanical signal transduction (Hamill, Paller & Jones, 2010). Lastly, locus 23486_75 was annotated to the hephaestin-like 1 (HEPHL1) gene that may function as a ferroxidase and may be involved in copper transport and homeostasis, while locus 33398_46 mapped to thioredoxin-related transmembrane protein 4 (TMX4) that may act as a reductase in the calnexin folding complex (Sugiura et al., 2010).

Table 1 Summary of outlier loci detected for American pika sampling sites along the Pyramid Peak (PP) and Thornton Lake (TL) elevational transects in North Cascades National Park.

Locus	SNP	Transect	F ST a	Top blast hit (accession)	Abbreviated description	
21404_70	C/T	PP	0.315	AC234826	Ochotona princeps clone VMRC40-45K5	
33398_46	C/T	PP	0.261	XM004593395	Ochotona princeps thioredoxin-related transmembrane protein 4	
57863_76	A/G	PP	0.255	NG008089.1	Homo sapiens receptor tyrosine kinase-like orphan receptor 2	
23902_347	C/T	TL	0.248	AC237024	Ochotona princeps clone VMRC40-172G4	
46878_140	C/G	TL	0.234	AC234901	Oryctolagus cuniculus clone 0087B06	
72966_67	A/G	PP	0.229	AL358859	Human DNA sequence from clone RP11-545G13 on chromosome 1	
59691_160	C/T	TL	0.223	AC234021	Ochotona princeps clone VMRC40-93N24	
23486_75	A/C	TL	0.209	XM004585173	Ochotona princeps hephaestin-like 1	
110148_49	A/G	TL	0.196	AC165118	Oryctolagus cuniculus clone 16788057J9	
94981_43	A/T	PP	0.194	AC236101	Ochotona princeps clone VMRC40-347J6	
43241_27	C/T	TL	0.171	AC233835	Ochotona princeps clone VMRC40-526O13	
108547_114	A/T	TL	0.165	XM004587540	Ochotona princeps platelet-derived growth factor receptor, beta	
87086_98	C/G	TL	0.155	XM004593191	Ochotona princeps putative uncharacterized protein FLJ46204-like	
28594_45	C/T	TL	0.153	NG007853	Homo sapiens laminin, alpha 3	
Notes.

a FST values significantly higher than under neutral expectations; averaged over populations.

Table 2 Genetic variation within American pika sites along the Pyramid Peak (PP) and Thornton Lake (TL) elevational transects in North Cascades National Park.

Site	Elevation	n	P	Ho	He	Ng	Fis	
PP1	450	8	0.774	0.282*	0.372	0.183	0.260*	
PP2	820	5	0.661	0.314*	0.425	0.213	0.295*	
PP3	1,330	5	0.837	0.403	0.403	0.329	0.001	
PP4	1,580	11	0.943	0.383	0.359	0.295	−0.071	
TL1	490	6	0.777	0.368*	0.400	0.202	0.088*	
TL2	780	10	0.839	0.339*	0.362	0.237	0.067*	
TL3	1,390	13	0.947	0.336	0.349	0.292	0.039*	
TL4	1,700	9	0.908	0.356	0.364	0.272	0.023*	
Notes.

Elevation in meters; sample size (n); proportion of polymorphic loci (P); observed heterozygosity (Ho); unbiased expected heterozygosity (He); gene diversity (Ng); inbreeding coefficient (Fis).

* p < 0.05.

Population genetic analyses

The proportion of polymorphic loci varied across the sampling sites, with the lower elevation sites (PP1, PP2, TL1) exhibiting substantially lower numbers (P = 0.661–0.777) than found at the mid- and high-elevation sites (P = 0.837–0.947) for both transects (Table 2). Similar trends were seen for gene diversity along both transects where the low and mid- elevation sites recorded the lowest values relative to higher elevation sites (Table 2). Indeed, both measures of site-level genetic variation were significantly correlated with elevation (P: r2 = 0.557, p = 0.034; Ng: r2 = 0.738 p = 0.006; Fig. 2). Although P significantly correlated with sample size (r2 = 0.635, p = 0.018), this was not the case for gene diversity (r2 = 0.0813, p = 0.493) or elevation (r2 = 0.184, p = 0.289). This general trend of increasing variation with elevation seemed to hold for observed heterozygosity along the PP transect, but were stable across TL sites (Ho: 0.336–0.368; Table 2). Yet, all low (PP1, TL1) and mid-low (PP2, TL2) sites exhibited significant, genome-wide evidence of heterozygote deficit. Interestingly, significant inbreeding was also detected at the low and mid-low elevation sites along PP, with no such evidence at the higher elevation sites (PP3, PP4; Table 2). All sites along the TL transect exhibited evidence of inbreeding (Table 2).

Figure 2 Elevational patterns of genetic diversity within American pika in the North Cascade National Park.

Solid line shows the correlation between proportions of polymorphic loci (P; circles) with elevation (r2 = 0.557 p = 0.034). Dashed line shows the correlation between gene diversity (Ng; squares) with elevation (r2 = 0.738 p = 0.006).

The AMOVA revealed that a significant amount of variation (p < 0.0001) was exhibited both among transect (4.14%, d.f. = 1) and among sites within transect (2.01%, d.f. = 6), with the remaining found within populations (93.05%, d.f. = 126). These patterns were congruent with those from pairwise θ estimates, with the highest values generally displayed by among transect comparisons, but where all comparisons were significant (Table 3).

Table 3 Pairwise θ estimates for American pika within and among the Pyramid Peak (PP) and Thornton Lake (TL) elevational transects in North Cascades National Park.

Site	PP1	PP2	PP3	PP4	TL1	TL2	TL3	TL4	
PP1	–	0.066	0.056	0.054	0.105	0.109	0.101	0.098	
PP2		–	0.040	0.049	0.096	0.101	0.087	0.088	
PP3			–	0.020	0.072	0.072	0.062	0.056	
PP4				–	0.065	0.068	0.059	0.053	
TL1					–	0.051	0.040	0.042	
TL2						–	0.027	0.030	
TL3							–	0.015	
TL4								–	
Notes.

Results based on 3,748 neutral single nucleotide polymorphisms. All pairwise θ estimates were significant (p < 0.05). Pairwise values for among transect comparisons shaded in gray.

The Bayesian clustering analyses based on 3,748 neutral loci revealed strong evidence for two clusters within the dataset (ΔK2 = 249.0), corresponding to the two transects (Fig. 1). When analyzing the PP transect separately, additional substructure (K = 2; Fig. 3) was found using neutral (ΔK2 = 33.1) and outlier (ΔK2 = 123.1) loci. In both cases, the low elevation site (PP1) represents a largely distinct cluster relative to all other sites. Similarly, substructure was found along the TL transect when using outlier loci, with strong evidence for two (ΔK2 = 473.3) and three (ΔK3 = 314.6) clusters. In the K = 2 plot, TL2 represented a distinct cluster, while in the K = 3 plot, the lower elevation sites (TL1, TL2) were each separate clusters relative to the high elevation sites. Although the Evanno, Regnaut & Goudet (2005) method would favor K = 2, the method described by Pritchard, Stephens & Donnelly (2000) for inferring the optimal number of clusters would suggest K = 3 given that ln Pr(X|K) clearly plateaus at this value (Table S1). No substructure was found along the TL transect based on neutral loci.

Figure 3 Structure bar plots depicting the model-based clustering results for Thornton Lake (TL) and Pyramid Peak (PP) sites based on outlier loci (above) and neutral loci (below).

Analyses for the TL transect revealed evidence for both K = 2 (ΔK = 473.3) and K = 3 (ΔK = 314.6; plot shown) based on 37 outlier loci, and K = 1 (K = 2 plot shown for display purposes) based on 3,748 neutral loci. Analyses for the PP transect revealed evidence for K = 2 (=123.1) based on 18 outlier loci, and K = 2 (ΔK = 33.1) based on 3,748 neutral loci.

Discussion

Conservation genomics has become an increasingly popular term in the literature, yet practical examples are limited (Shafer et al., 2015), including explicit consideration of the efficacy of genomic data collection from non-invasively collected starting materials. Here, we demonstrated the ability to identify 3,803 high confidence SNPs and recover genotypic data from low quantity DNA originating from non-invasively collected American pika hair samples. These data allowed us to detect outlier loci across elevational transects, identifying several candidate gene regions that exhibit putative signatures of divergent selection and that can be investigated in future studies for formulating mechanistic hypotheses. Moreover, the broad-scale genomic coverage enabled precise estimation of population-level parameters, including standard diversity indices, inbreeding, and structure within and among sampled transects.

We found genetic variation to be significantly correlated with elevation (Fig. 2), with sites at the lower fringe of American pika distribution in North Cascades National Park exhibiting substantially lower levels of gene diversity. No such associations were found in a previous microsatellite-based study conducted across elevationally-distributed sites in British Columbia, Canada (Henry, Sim & Russello, 2012). The detection of significant genome-wide evidence of heterozygote deficit at low elevation sites in both transects further suggests inbreeding may be leading to the observed patterns (Table 2), a particular concern for PP1, TL1 and TL2 given their apparent distinctiveness from higher elevation sites (Fig. 3). Due to their specific habitat requirements, patchy distribution, and life history, American pikas were long thought to regularly interbreed with close relatives based on observational studies (Smith & Ivins, 1983). Yet, molecular marker based studies have altered our understanding of American pika breeding behavior, revealing evidence for mate choice based on intermediate relatedness in one case (Peacock & Smith, 1997b), while another found no evidence for inbreeding across elevationally-distributed sites (Henry, Sim & Russello, 2012). This latter study conducted in Tweedsmuir South Provincial Park, British Columbia, Canada also found evidence for broad-scale and fine-scale population structure, detecting restricted gene flow among transects as well as among sites within transect and potentially driven by climatic factors (Henry, Sim & Russello, 2012). Although conducted at a different scale, Castillo et al. (2014) found a high degree of connectivity among geographically proximate sites in Crater Lake National Park, Oregon, USA, but restricted gene flow at a broader scale likely driven by topographic complexity and water. Here, we detected similar coarse-level patterns, detecting strong population genetic structure across transects but some evidence of connectivity between sites within transects in North Cascades National Park. It is worth noting that the three studies spanned the distribution of the Cascades lineage of American pika (Galbreath, Hafner & Zamudio, 2009), conducted in the north (Henry, Sim & Russello, 2012), south (Castillo et al., 2014) and central (this study) portions of the range. Given that pikas are considered by some to be sentinels of climate change (Hafner, 1993; Smith, 1974), further investigation is warranted to infer underlying mechanisms associated with dispersal ability in pikas that may be further enhanced by comparative analyses of elevationally- and latitudinally-distributed sites.

More broadly, our results highlight a range of issues that must be considered when pairing genomic data collection with non-invasive sampling. First, sampling protocols must endeavor to minimize non-target DNA during the collection process. In our case, the use of tape-based, non-invasive hair snares allowed us to collect genomic data from 67 individuals of American pika, but also yielded 19 samples that were almost entirely composed of DNA sequence reads from non-target organisms, primarily other small mammals that were likely using the same talus habitat. Precautions to avoid such contamination will vary according to the methods in which samples are obtained, but are of critical importance given the non-targeted nature of NGS approaches. Depending upon study questions, the use of exon-capture or other approaches for preferentially targeting the DNA of study organisms within a mixed sample may help to minimize contamination and maximize cost-effectiveness of downstream sequence data (Avila-Arcos et al., 2011; Carpenter et al., 2013; Good, 2011). Exon-capture, in particular, has been effectively applied to historical DNA collected from museum specimens (Bi et al., 2013), which typically yield DNA of lower quantity and quality similar to non-invasively collected starting materials. Yet, these approaches are substantially more costly and, in the case of exon-capture, limited to expressed regions of the genome. However, for some non-invasively collected DNA such as feces, the use of capture approaches may be obligatory (Perry et al., 2010).

Additionally, rigorous assessments of resulting DNA sequence data must be undertaken to ensure quality control. In the current study, we used a genotyping-by-sequencing approach for reduced representation genomic data collection. We had the advantage of publicly available American pika genomic scaffolds that allowed us to initially filter our data based on SNPs assembling to these references. At present, such resources may not be available for many organisms of conservation interest. In such cases, we recommend using the closest available genome to inform reference assembly of identified SNPs (in our case, this would have been the European rabbit; Lindblad-Toh et al., 2011). If no suitable reference genome is available, investigators may want to consider capture approaches for genomic data collection (as discussed above).

Analytical frameworks must also be carefully considered in relation to recovered sequence coverage depth in studies using non-invasively collected samples. In our case, we used explicit parameters related to coverage and amounts of allowable missing data to confidently reconstruct genotypes from our sampled individuals. While there is no clear standard in the literature, choice of such parameters is a balance between maximizing the number of loci and minimizing null alleles when reconstructing genotypes. Yet, reconstructed genotypes may not be necessary for all study questions, especially those primarily focused on estimating population-level parameters rather than individual-based measures (e.g., admixture coefficients, individual identification, parentage probabilities; but see Buerkle & Gompert, 2013). In such cases, low density genomic scans based on more individuals or sites in the genome may provide highly accurate and precise population parameter estimates, even at as low as 1X coverage (Buerkle & Gompert, 2013; Fumagalli, 2013). Analytical pipelines continue to be developed that implement population genetic analysis methods that account for the statistical uncertainty of NGS data (Fumagalli et al., 2014), with empirical examples now found in the literature (Cahill et al., 2013).

Overall, NGS data and population genomic analyses hold great promise for informing conservation-related studies, substantially increasing the number of markers to allow for more accurate and precise estimates of population structure and demographic parameters (Primmer, 2009), as well as the ability to detect adaptive genetic variation for informing conservation unit delimitation (Funk et al., 2012) and decision frameworks aimed at reducing the long-term impacts of climate change on biodiversity (Hoffmann et al., 2015). Here, we have shown that with careful consideration, genomic data collection is compatible with the non-invasive sampling required in practice for many conservation-related studies.

Supplemental Information

Table S1 Summarized Structure output including ΔK for the Thornton Lake elevational transect based on 37 outlier loci

Click here for additional data file.

We thank Regina Rochefort, Roger Christophersen, Liesl Erb, Kelsey Robson, Shane Schoolman, and Aidan Beers for assistance with site selection and sampling. Evelyn Jensen and Andrew Veale provided feedback on the manuscript. Holly Buhler helped prepare the map figure. A special thanks to the Department of Organismic and Evolutionary Biology at Harvard University for hosting MR during his sabbatical visit, during which the initial development of this work occurred.

Additional Information and Declarations

Competing Interests

Author Contributions

Animal Ethics

DNA Deposition

Paul D. Etter is the Director of Research and Eric A. Johnson is the founder of SNPsaurus, an organization that offers commercial nextRAD sequencing services. This organization provided no funding for this project.

Michael A. Russello conceived and designed the experiments, analyzed the data, contributed reagents/materials/analysis tools, wrote the paper, prepared figures and/or tables, reviewed drafts of the paper.

Matthew D. Waterhouse performed the experiments, prepared figures and/or tables, reviewed drafts of the paper.

Paul D. Etter and Eric A. Johnson analyzed the data, contributed reagents/materials/analysis tools, reviewed drafts of the paper.

The following information was supplied relating to ethical approvals (i.e., approving body and any reference numbers):

All samples were collected under United States Department of Interior National Park Service permit # NOCA-2014-SCI-0022 and in accordance with animal care protocol (A11-0371) as approved by the University of British Columbia’s Animal Care & Biosafety Committee.

The following information was supplied regarding the deposition of DNA:

Dryad: 10.5061/dryad.61691.

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
