# Peer review of "From promise to practice: pairing non-invasive sampling with genomics in conservation"

_PeerJ, doi:10.7717/peerj.1106_

## Round 0.1 · original submission · Minor Revisions

Like the reviewers I liked the overall premise of the study but there seems to be significant grounds for improvement of the ms regarding focus and clarity. The aims seems to be primarily technical so more detailed methodology is needed. Also a better comparison with alternative approaches is called for, both for non-invasive sampling and genetic analyses. Please refer to the reviewers' comments for detailed requests.

Reviewer 1 ·

Basic reporting

In this study the authors address a problem in the field of conservation genomics: the need to collect samples non-invasively does not always yield enough DNA for high-powered NGS methodologies. Both the problem and their planned approach are presented nicely and the paper is overall well-written. I also thought the figures and tables were chosen well to support their message.

Experimental design

I believe (and the authors may correct me if I am wrong) that this paper is meant to be more of a 'proof of concept' type paper (re: non invasive sampling and NGS) than a study on pika genetics. Therefore, I believe it would be strengthened by some additional explanation of the methodology chosen and their evaluation of its success: for example, how were the 9bp selective sequences chosen, and were any attempts made to examine how well the loci they obtained from this method represent the entire genome?

Validity of the findings

The quality control methods used to filter samples and loci appear sound and well considered. Sample sizes are small although for a concept study this is not a big deal. I thought that they actually sell themselves a bit short in their conclusions and could be a little more overt in stating the advantages of their approach relative to other methods like exon capture.

Additional comments

Specific comments

95 ‘RAD-seq’ was mentioned earlier in the introduction, and ‘nextRAD’ here. These are two different approaches (and in fact the RAD stands for different things). Some clarification would be helpful for those who are new to these methodologies.

106-107 Does ‘this area’ refer to the transects, North Cascades Park, the Pacific northwest, etc?

107 typo – should be ‘affected’

113 I am assuming 15m is outside the dispersal range of pika?

114-115 How was this tested? What was the outcome? (i.e. conclusive determination that no individual was sampled more than once, or some kind of likelihood)

130 How were the 9 nucleotides chosen? Why 9 (not 8, not 10)? Was there a mathematical procedure or simply trial and error to determine this sequence would select a good number of potential loci? Additional explanation would improve the ms especially since ‘nextRAD’ seems to be a newer method and potentially of interest to a lot of readers.

143-147 Does ‘population’ here refer to all of the reads within a single individual, or to the combined set of reads across all individuals examined? I am inclined to draw the latter meaning from the wording used, but hesitate at the phrase ‘more than the expected maximum of two alleles.’ Within an individual, two would be expected, but across an entire population there could easily be loci with more than two. A little clarification would aid understanding.

218 It would be useful to refer to Table 2 here – so the reader can keep the distribution of sample sizes in mind going forward.

230 typo – ‘functions’ not ‘function’

Table 1 For those unfamiliar with outlier detection using Bayescan it would be helpful to indicate in the table legend or a footnote what these FST values mean (as opposed to traditional FSTs): basically, if I am understanding correctly, they correspond to how strongly the locus itself deviates from neutrality (so that neutral loci would have FST values of 0)?

Table 2 The higher elevation sites seem to have more individuals sampled on average. I am curious if the authors explored whether this could have contributed to the gradient in P, for example? (Perhaps by including n as an explanatory variable in the regression?)

280 Do the 3803 SNPs spread out evenly across chromosomes? Are some areas more vs. less represented, for example due to repetitive sequence?

282-287 Are there particular traits exhibited by pikas that could explain these findings? (e.g. social structure, dependence on heat-intolerant food sources). I know that this isn't the point of the paper, but perhaps the authors could refer to a key paper(s) containing this information for readers that are interested.

299-310 This discussion begs the question of why a conservation geneticist would choose to do nextRAD if exon-capture can also be used with low amounts of starting material, eliminates the problem of non-target contamination, and doesn’t require a reference genome. I suspect the answer is time and cost, but it would strengthen the authors’ assertions about the utility of their approach to state this more overtly.

315 ‘such of parameters’ – eliminate ‘of’

Reviewer 2 ·

Basic reporting

No comments

Experimental design

Paragraph (Line 84-94)
The authors mention the use of AFLP’s in some of their previous studies and list their undesirable properties but there was no discussion about how much better the nextRAD technology performs compared with the old AFLP or microsatellite data. Is there comparable data generated with microsatellites or AFLP’s from the same localities?

Methods (Line 111-117)
Also, most validation studies for non-invasive samples use some kind of tissue or blood sample as a control to measure efficacy and reliability of the low quantity/quality DNA sample. Why was there not a control sample or samples with blood or tissue vs. hair comparison in this study?
This would also give the reader a sense of how much allelic drop-out and genotyping error there is with hair samples.

DNA isolation section (Line 119-123)
There were no specifications given as to any precautions taken to avoid contamination during lab procedures such as use of negative controls and any other precautions taken during DNA extraction of samples to avoid contamination?

Population genetics metrics (Lines 284-287)
How do the population genetics metrics compare to results of their previous Pika studies obtained usin AFLP's or microsatellites?

(Line 214-215) I was confused about this statement “Three samples had high bacterial contamination (Spermophilus; TL1A13b, PP3A02a, 215 PP1A24a) as evidenced by the number of matching sequence reads (15-22%)”.
Spermophilus is a ground squirrel not a Pika or bacteria?

Validity of the findings

The authors evaluated the feasibility of using nextRAD genotyping-by-sequencing of noninvasively collected hair samples to simultaneously identify and genotype SNPs in American pikas from two transect along an elevational gradient in the North Cascades National Park. They highlight a range of issues that must be considered when pairing genomic data collection with non-invasive hair sampling, and analytical approaches for maximizing cost-effectiveness and information content of recovered genomic data. The only concerns I had were that this paper really just addressed the problems associated with noninvasive hair sampling and the results are not really applicable to the other more commonly used method of scat (feces) for noninvasive studies. Therefore, I suggest that the authors make this clear in their title, their introduction and discussion and perhaps refer the reader to other approaches used for capturing and sequencing of endogeneous DNA from fecal samples. They do mention that “exon-capture, in particular, has been effectively applied to historical DNA collected from museum specimens (Bi et al.
2013), which typically yield DNA of lower quantity and quality similar to non-invasively collected starting materials” but I think that they need to clarify that this capture approaches are necessary for fecal samples:
Perry, G. H., Marioni, J. C., Melsted, P., & Gilad, Y. (2010). Genomic-scale capture and sequencing of endogenous DNA from feces. Molecular Ecology, 19(24), 5332–5344. doi:10.1111/j.1365-294X.2010.04888.x

Also, I think this argument can be substantiated by referencing the below paper that actually does an exhaustive survey of different types of methods and more recent than the Waits & Paetkau 2005 that they cite:
Beja-Pereira, A., R. Oliveira, P.C. Alves, M.K. Schwartz, and G. Luikart. 2009. Advancing ecological understandings through technological transformations in noninvasive genetics. Molecular Ecology Resources 9:1279–1301.

Additional comments

Other topics that were not discussed:

In their outlier detection analysis they identified 37 loci along the TL transect and 18 loci along the PP transect and none of which were shared. Why don’t they share outlier loci? This should be briefly clarified in the discussion.

The proportion of polymorphic loci varied across the sampling sites, with the lower elevation sites (PP1, PP2, TL1) exhibiting substantially lower numbers than found at the mid- and high-elevation sites for both transects. Any speculation as to why the lower elevation sites are more inbred? What do the results obtained in previous Pika studies with microsats or AFLP’s reveal in terms of genetic diversity in lower vs higher elevation?
Also, what differences do they find in terms of fine-scale structure resolution with SNP’s vs. AFLP’s or microsats?

---

## Round 0.2 · accepted · Accept

Thank you for addressing the reviewers comments carefully and thoroughly. I am delighted to accept this study that shows the potential usefulness of high throughput NextGen sequencing of even degraded samples.

Reviewer 1 ·

Basic reporting

No comments

Experimental design

No comments

Validity of the findings

No comments

Additional comments

The authors have done a great job addressing my comments and I have nothing further to add.

Reviewer 2 ·

Basic reporting

No comments

Experimental design

No comments

Validity of the findings

No comments

Additional comments

The authors have adequately addressed my concerns from my previous review and I am satisfied with their response to my comments.